# Estrobolome and Hepatocellular Adenomas—Connecting the Dots of the Gut Microbial β-Glucuronidase Pathway as a Metabolic Link

**DOI:** 10.3390/ijms242216034

**Published:** 2023-11-07

**Authors:** Sandica Bucurica, Mihaela Lupanciuc, Florentina Ionita-Radu, Ion Stefan, Alice Elena Munteanu, Daniela Anghel, Mariana Jinga, Elena Laura Gaman

**Affiliations:** 1Department of Gastroenterology, “Carol Davila” University of Medicine and Pharmacy Bucharest, 020021 Bucharest, Romania; sandica.bucurica@umfcd.ro; 2Department of Gastroenterology, “Dr. Carol Davila” Central Military Emergency University Hospital, 010242 Bucharest, Romania; mihaelalupanciucmihail@gmail.com; 3Department of Infectious Diseases, “Dr. Carol Davila” Central Military Emergency University Hospital, 010242 Bucharest, Romania; dr.stefanion@gmail.com; 4Department of Medico-Surgical and Prophylactic Disciplines, Titu Maiorescu University, 031593 Bucharest, Romania; dralicepopescu@yahoo.com (A.E.M.); drdanielaanghel@gmail.com (D.A.); 5Department of Cardiology, “Dr. Carol Davila” Central Military Emergency University Hospital, 010242 Bucharest, Romania; 6Department of Internal Medicine, “Dr. Carol Davila” Central Military Emergency University Hospital, 010242 Bucharest, Romania; 7Department of Biochemistry, “Carol Davila” University of Medicine and Pharmacy Bucharest, 020021 Bucharest, Romania; laura.gaman@umfcd.ro

**Keywords:** hepatic adenoma, hepatocellular adenoma, gut microbiota, estrogen, GUS, β-glucuronidase enzymes, liver, fatty liver, steatohepatitis, HCA

## Abstract

Hepatocellular adenomas are benign endothelial tumors of the liver, mostly associated with female individual users of estrogen-containing medications. However, the precise factors underlying the selective development of hepatic adenomas in certain females remain elusive. Additionally, the conventional profile of individuals prone to hepatic adenoma is changing. Notably, male patients exhibit a higher risk of malignant progression of hepatocellular adenomas, and there are instances where hepatic adenomas have no identifiable cause. In this paper, we theorize the role of the human gastrointestinal microbiota, specifically, of bacterial species producing β-glucuronidase enzymes, in the development of hepatic adenomas through the estrogen recycling pathway. Furthermore, we aim to address some of the existing gaps in our knowledge of pathophysiological pathways which are not yet subject to research or need to be studied further. As microbial β-glucuronidases proteins recycle estrogen and facilitate the conversion of inactive estrogen into its active form, this process results in elevated levels of unbound plasmatic estrogen, leading to extended exposure to estrogen. We suggest that an imbalance in the estrobolome could contribute to sex hormone disease evolution and, consequently, to the advancement of hepatocellular adenomas, which are estrogen related.

## 1. Introduction

The spectrum of sex-hormone-responsive diseases or estrogen-related diseases is wide, from the well-known to the newly added, including but not limited to breast, ovarian, and prostate neoplasia, endometriosis, polycystic ovarian syndrome, multiple sclerosis, cancer of the thyroid and pituitary glands, schizophrenia, and obesity [1].

The mechanism of action of estrogen is complex and may be described as pleiotropic rather than just hormonal; in this regard, it has not yet been completely elucidated [1]. The concept of the “estrobolome” has been proposed, referring to the collective set of enteric bacterial genes which encode the products which are capable of metabolizing, reactivating, conjugating, and reabsorbing free estrogen [2,3,4]. It is known that the estrobolome plays an important role in the human endocrine system; specifically relevant in the metabolization of estrogen are bacterial species producing β-glucuronidase (GUS) enzymes [2,4,5,6,7]. Gut microbial β-glucuronidase (gm-GUS) is the most studied among such bacterial phyla; it is responsible for supporting the deconjugation of conjugated estrogen and encouraging the resorption of free estrogen [2,4,5,8].

Hepatocellular adenoma (HCA) or hepatic adenoma is defined as a benign liver lesion, seen in patients with excessive exposure to estrogen [9,10] (the higher the dose of estrogen therapy, the higher the risk of HCA), genetic and metabolic syndromes or who have undergone anabolic androgen therapy [9,11,12,13,14]. Very few cohort studies and no epidemiological studies have argued that reductions in estrogen doses can reduce the risk of HCA. However, while the HCA incidence rate in the general population is about 1:1,000,000 per year, long-term users of oral contraceptive pills are 30% more likely to develop HCA [9,11].

Since the development of molecular techniques, notably including metagenomic techniques, our knowledge of the human gastrointestinal (GI) microbiome has expanded considerably. We possess data that show that the gut microbiota (bacteria, viruses, Archaea, and Eukaryotes [2,15]) is diverse and contributes to intestinal permeability, digestion, metabolism, and immune responses and, in general, influences the host’s health and disease occurrence [2,16].

The metabolic potential of the human gut microbiota is enormous, being intensely connected with human physiology; for example a vast number of enzymes are implicated in numerous metabolic pathways (e.g., the production of bioactive peptides such as branched-chain amino acids, short-chain fatty acids, neurotransmitters, intestinal hormones), the biosynthesis of vitamins (ex. thiamine, folate, biotin, riboflavin, pantothenic acid, half of the daily required vitamin K), and secondary bile acid conversion [8].

Although research in this field is limited at present, it is known that various features can influence the status of the GI microbiome, like diet, race, age, sex, antibiotic use, environment, and psychological factors [5]. It is plausible to assume that the sex of the host (based on their reproductive system, chromosome type/hormones) plays an important role in the functionality of the gut microbiome [17,18]. Sender et al. reported that the number of cells in the gut microbiota is similar to that of human cells in the body; additionally, the ratio (bacterial cells to human cells) is different from one sex to another, being roughly 1.3:1 for males and 2.2:1 for females [19]. Various cohort studies in different years (in France [20,21], Denmark [20], Germany [20,21], the Netherlands [20,22], UK [20], China [23,24], USA [25,26], Spain [17], Italy [21,27], Japan [28], and Sweden [21]), even if they included a large number of variables, did not study the sex differences of the gut microbiota precisely. The results of these studies stated that the differences in microbial rate, number, and characteristics between sexes are unreliable or need to be studied more [29,30,31]. Haro et al. examined whether the intestinal microbiota is influenced by gender and Body Mass Index (BMI) [22]. Their study noted that even with all the observations, sex explained only 0.5% of the total variation in the gut and this percentage was influenced by the BMI [17]. Also, animal studies concluded that a high BMI is associated with an increase in the Firmicutes/Bacteroidetes ratio; however, studies in humans have resulted in conflicting conclusions [31,32]. A possible explanation for this is the inter-individual heterogeneity of gut microbiota exposure [17]. In one report, the sex differences in the gut microbiota became relevant when enteric infection was present [26]. Patients with enteric infections (*Salmonella*, *Shigella*, *Shiga* toxin-producing *Escherichia coli*, *Campylobacter*) had a minor abundance of *Bacteroides* (in females) and *Escherichia* (in males); this difference was not detected in healthy individuals [31]. Sex hormones may be involved in these gut differences, as until puberty, it appears that there are no sex differences in the GI microbiome [5].

This process is particularly relevant in oncology, gynecology, and gastroenterology, potentially explaining the contribution of GUS enzymes and the excessive reabsorption of unconjugated estrogen through enterohepatic circulation, leading to high concentrations of unbound estrogen in plasma and tissues. Being a critical participant in this pathophysiological process, extended exposure to estrogen is known to determine sex hormone-responsive diseases. Accordingly, high exposure to estrogen contributes to a variety of hormonal disorders, including endometriosis, ovarian cancer, endometrial cancer, breast cancer, and endometrial hyperplasia.

## 2. Methodology

We conducted a literature search using the following keywords: “hepatocellular adenomas”, “incidence of hepatocellular adenoma”, “classification of hepatocellular adenomas”, “gut microbiota”, “sex difference and gut microbiota”, “b-glucuronidases in adenomas”, “estrogen”, “estrobolome”, “estrogen and the gut-liver-brain-axis”, “estrogen and breast cancer”, “etiology of hepatocellular neoplasm”, and “malignancy of hepatocellular adenoma” in the PubMed, MEDLINE, ScienceDirect, and National Library of Medicine databases. For this review, articles relevant to the topic written in English were selected. Articles with no full text availability were excluded. The primary purpose of our extensive review was to highlight the missing links between the gut microbiota, estrobolome, and the etiopathogenesis of HCAs.

## 3. Hepatocellular Adenomas

The frequency ratio of HCA is 8:1 in females and males; this can be explained by the fact that exogenous estrogen therapy (OC) is usually prescribed for women [12]. Other circumstances linked with HCA development include glycogen storage disease type I and type III (GSD) [9,11,14], hepatocyte nuclear factor 1α maturity-onset diabetes in young people (HNF1α-MODY) [33], history of liver diseases (non-alcoholic steatohepatitis, vascular diseases, alcoholic cirrhosis) [33], galactosemia, tyrosinemia, familial polyposis coli, polycystic ovary syndrome [7], and β-thalassemia [12,34] (Table 1).

However, 75% of hepatocellular carcinoma cases occur in males [35]. Differential diagnoses between HCA, well-differentiated HCC, and focal nodular hyperplasia (FNH) can be difficult, especially in males [14,36].

The clinical guidelines of the European Association for the Study of the Liver (EASL) for the treatment of benign liver tumors [36], published in 2016, state that MRI is the best imagistic investigation option, offering the opportunity to differentiate HCA in up to 80% of the cases. MRI has the capacity to identify 90% of HNF-1α HCA (H-HCA) or Inflammatory HCA (I-HCA) cases [11,36]. In comparison, the identification of β-catenin-activated HCA or unclassified HCA is nearly impossible with MRI, although immunohistochemistry (IHC) can provide a definite HCA subtype in 2/3 of cases [36]. The EASL guide also notes that the treatment of HCA needs to be based on sex, size, and pattern of progression; as a result, initial conservative treatment for women involves the discontinuation of OC plus weight loss and 6 months of observation after the lifestyle changes have been made [11,36].

**Table 1 ijms-24-16034-t001:** Genotype–phenotype classification of hepatocellular adenomas [5,12,33,34,37,38].

HCA Classification	Incidence	Risk Factors	Clinical Characteristics	Complications
HNF1α-HCA (H-HCA)	30–35%	Somatic or germline mutations of HNF1α; Over-exposure to exogenous estrogen [14]	Female; Familial adenomatosis; Fatty liver disease MODY 3 diabetes;	HCA Ø < 5 cm lower chance of malignancy; Lowest malignancy potential vs. other HCA subtypes; Rarely bleeding;
Inflammatory-HCA (I-HCA)	35–45% the majority cases of HCA	Deletions of IL6ST gene; Mutations of IL-6/JAK/STAT2 family; Over-exposure to exogenous estrogen [14]; Metabolic syndrome and, or Obesity [14], Alcoholism	Obesity; Inflammatory phenotype: appearance of high serum amyloid (SAA) and C-reactive protein (CRP);	Rarely or not associated with malignancy potential;
Sonic hedgehog HCA	4%	Over-exposure to exogenous estrogen [14]; Metabolic syndrome and, or Obesity [14]	Obesity	Greater possibility of bleeding
Unclassified HCA	5–10%	-	-	-
**β1-catenin (cadherin-associated protein) HCA** [35]				
β-catenin HCA exon 3 [33]	10–15%	Mutation in CTNNB1 gene exon [3]; High blood androgen exposure [14]; Liver vascular disease	Often in males [10]; Younger individuals; Frequently isolated tumors;	Highest chance of Hepatocellular carcinoma (HCC) [10]; Potential of bleeding [37];
β-catenin HCA exon 7–8 [33]	5–10%	Over-exposure to exogenous estrogen [14]	Often one tumor; Younger individuals; Inflammatory phenotype;	Low risk of HCC; Greater possibility of bleeding
**Rare HCA subtypes** [33]				
Pigmented HCA	very rare	Various genetic mutations, with pigment deposition of lipofuscin		High-level chance of carcinogenesis
Myxoid HCA	Rare variant of H-HCA with extra mutations/separate subtype of HCA	Loss of L-FABP gene expression and/or HNF-1 α mutation; Recurrent mutations in PKA regulation of pathway (GNAS, CDKN1B and RNF123) or mutations of PKA pathway;	Older age	High-level chance of carcinogenesis
Atypical HCA/Borderline HCA/HUMP	-	-	Individuals > 50 years	Need more studies
I-HCA in cirrhotic liver	Need more studies	Fatty liver disease; Liver vascular disease; Metabolic syndrome; Alcoholic cirrhosis	-	Need more studies

HUMP—hepatocellular neoplasm of uncertain malignant potential; MODY 3 diabetes—maturity-onset diabetes of the young type 3; PKA—protein kinase A; CTNNB1—cadherin-associated protein B1.

If the lesion/lesions are <5 cm, a reassessment will be made after one year; if the nodule/nodules are greater than 5 cm in size and continue to grow, surgical removal is advised [36]. Regardless of HCA size, in males or in any instance of proven β-catenin mutation, HCA resection is recommended [36]. HCA is associated with a risk of hemorrhage (10–20%) in larger tumors (>5 cm) and, rarely, a risk of malignant transformation (<5%) into hepatocellular carcinoma (HCC) [10,35]. A retrospective cohort study published in 2019 in Liver International described the effect of oral contraceptive pill cessation on hepatocellular adenoma diameter. The study stated that 98% of the observed HCA cases remained unchanged or regressed after cessation of OC [39]. Of 267 patients with HCA, 78 were OC-HCA patients. At a median time of 1.3 years after OC interruption, 37.2% (29 HCA) showed ≥30% regression, 5.1% (4 HCA) showed complete regression, 56.4% (44 HCA) remained unchanged, and 1.3% (1 HCA) had progressed [39]. In another report that described HCA regression rates and the timing of HCA resection, it was reported that 15% of HCA participants in the study displayed regression ≤ 5 cm at a six-month follow-up; also, larger HCA regularly needed more than six months to regress to a diameter < 5 cm [40]. Male sex represents a high risk factor for the oncogenesis of HCA, alongside the β-catenin exon 3 mutations. There are substantial diagnostic difficulties for smaller lesions. Magnetic resonance exams with hepatocyte-specific contrast agents (such as gadoxic acid) were indicated to identify early-stage malignancies [41].

## 4. Estrogen Receptors and Evolution to Malignancy

From a pathological point of view, HCAs are well-differentiated tumors with non-atypia cytology appearance. However, the nuclei could be slightly enlarged but with minimal modification of the nuclei–cytoplasm ratio. HCAs were determined to feature preserved reticulin and Ki-67 proliferative activity of less than 2% and uneven CD34 staining [42]. Up to 50% of HCAs are inflammatory adenomas, while HNF1-α-inhibited adenomas represent more than a third of all HCA cases [42]. Hepatic steatosis was mostly associated with inflammatory adenomas, with positivity for serum amyloid A or C-reactive protein, and the genetic background varied, with miscellaneous mutations of the genes that activate the JAK/STAT pathway like JAK1, GNAS, STAT3, or IL-6ST [42].

Sigma receptor 1 (Sig-R1) represents a membrane protein that belongs to the endoplasmic reticulum as a component of the mitochondria-associated membrane (MAM) region [43]. It is linked to one of the multifunctional characteristics of the endoplasmic reticulum, which is a response to stress with neuronal protective activity, while Sig-R1 balances the production of reactive oxygen species at the mitochondrial level [44].

Although Sig-R1 is pervasive, it is mostly expressed in hepatic tissue and in the central nervous system, and has neuroactive steroid ligands (like progesterone and pregnenolone), even though its role has not been well-defined. Sig-R1 was found to be overexpressed by exposure to estrogen on α estrogen receptors (ER-α) [43]. ER-α are nuclear transcription factors that are activated by ligands, and when the complex with estrogen is formed, it acts on the genetic expression [45].

In preclinical studies, estrogen appeared to regulate, through direct or indirect pathways, a variety of hepatic proteins via estrogen receptors that are highly expressed in liver tissue. Additionally, previous studies have found that estrogen also participates in the proliferative process of hepatocytes [46]. This estrogen activity is ER-α mediated at the transcriptomic stage. This receptor is the target for estrogen signaling, and liganded ER-α exerts control over the transcription of genes that are specific to its binding [46]. A preclinical study by Guillaume et al. described the degree of liver ER-α involvement in metabolic balancesince ER-α was considered a biomolecule of interest for liver steatosis. That study concluded that the specific regulation of these hepatic estrogen receptors could prevent dysmetabolic disorders [47]. Consistently, Sig-R1 was found to participate in fatty liver disease and may modify the hepatocellular adenoma phenotype, since it is also a regulator of hepatic cell proliferation [43].

The potential pathways of HCA evolution to malignancy involve JAK-STAT pathway modulation and activation. In animal studies, the inhibition of cyclin D and estrogen receptor α36 (ERα36) super-expression were described [48]. In a study by Lau et al., a new connection between endoplasmic reticulum stress and impaired cyclin D and ER-α36 was suggested in the evolution of HCA to malignancy. Additionally, in HCA, it was found that inhibited activity of HNF1-α was associated with an overexpression of Sigma receptor 1; a possible explanation for this was that HCA could evolve from a different cellular line than HCC [43].

In order to compare HNF1-α HCA with normal hepatic tissue characteristics, Pelletier et al. identified more than 300 under-expressed and over 200 over-expressed genes in HNF1α hepatic adenomas [49]. The mutation leading to HFN1-α inactivation that is present in this type of hepatocellular adenomas was also described in some cases of hepatocarcinoma in non-cirrhotic livers; this led to the idea that this is a premature genetic alteration in hepatocellular carcinogenesis [49]. In the same study, this genetic mutation was associated with the inability of HNF1-α HCA to deactivate active estradiol, with a lipogenic, proliferative, and neoangiogenic effect [49].

## 5. Gastrointestinal Microbiome, Estrobolome and Bacterial Species Possessing β-Glucuronidase Enzymes

In 2011, the notion was proposed of the “estrobolome”, i.e., the total of the enteric bacterial genes that codify the products capable of metabolizing or reactivating conjugated estrogens or effecting the reabsorption of free estrogen [2,3,4]. In theory, the small intestine (specifically, the jejunum and ileum) is probably at the origin of GUS GI enzymatic activities [5]. Many bacterial genera and species in the human gut microbiota contain genes encoding β-glucuronidase. Sixty bacterial genera that colonize the human intestinal tract are known to encode β-glucuronidase and/or β-galactosidase [2].

In a comparative study, it was established that estradiol administrated orally or intravenously to young females could be converted into estradiol glucuronide. This metabolite can be measured in the blood and urine [5,32]. Interestingly, the amount of estrogen glucuronide found in urine samples was half of that found in the blood samples. This led to the belief that the digestive system is a significant place for estrogen glucuronidation [5,50].

While the GI microbiome can be moderated by estrogen, the gut itself also influences estrogen levels [7]. The GI microbiota can influence hormonal balance via estrobolome [2,3,4,5,7].

Bacteroidetes and Firmicutes are dominant in the GI tract and are responsible for encoding the primary source of the GI GUS [5,7,16]. The Gm-GUS function is regulated by two genes, Gus and BG. Both of these genes are represented in Firmicutes, and BG is expressed only in Bacteroidetes [5]. The Gus gene is the main gene responsible for the primary response of GUS enzyme activities, as species that carry the BG gene show low GUS enzyme activity [2,5]. One study reported that the main gm-GUS protein-producing species isolated from human fecal samples were Firmicutes [6], which were accordingly divided into six structural classes based on active sites: loop 1, mini-loop 1, loop 2, mini-loop 2, mini-loop 1,2, and no-loop [4,5]. These structural differences may be an indicator of substrate partiality among species. Firmicutes are primarily loop 1, mini-loop 1, and no-loop types; in contrast, *Bacteroides* GUS are not loop 1 type [5]. Another type of gm-GUS that displayed the exclusive ability of small-molecule glucuronide cleavage is FMN binding [5]. One study showed that the mini-loop 1, loop 2, and no-loop enzymes, even if they were able to process preferential polysaccharides, could cleave the smaller glucuronide substrate very well [16]. Therefore, it was proposed that Firmicutes gm-GUS holding loop 1, mini-loop 1, and FMN structures are the incontestable source of estrogen-reactive gm-GUS [5].

β-glucuronidase is a glycosyl hydrolase that can catalyze the hydrolysis of the O- or S-glycosidic moieties and free the aglycones from glycosides.

Metabolized into glucuronide in the liver, estrogen-conjugated metabolites are excreted through bile, renal clearance, and feces [2,5].

Estradiol and estrone are lipophilic but with some water solubility properties; the sulfate and glucuronide estrogen forms have a high water solubility. Deconjugated estrogens from the gut are reabsorbed into the enterohepatic circulation by easily diffusing through cell membranes [51,52,53]. Studies have found that a substantial amount of estrogen metabolites enter the GI tract by the gm-GUS for deconjugation. GUS proteins are capable of stopping the process of inactivation and excretion by cleaving glucuronic acid from estrogen glucuronides. As a consequence, biologically inactive estrogen becomes biologically active and can be reabsorbed through enterohepatic circulation and move throughout the body. Gm-GUS can modulate the entero-hepatic flow and the reabsorption of free estrogens. This emphasizes its important role in estrogen metabolism [5].

In a preclinical study, Zhong et al. demonstrated that the pharmacokinetic behavior of the non-steroidal anti-inflammatory drug (NSAID) diclofenac was modified after antibiotic interventions due to bacterial β-glucuronidase inhibited activity in the intestines (after the administration of ciprofloxacin). In that research, reduced β-glucuronidase activity was observed, particularly in the ileocecal segment. The decreased plasmatic concentration of diclofenac was attributed to the elimination of the enterohepatic recirculation of diclofenac, highlighting the absence of NSAID enteropathy when an antibiotic was administered [54]. This is the most frequently observed mechanism leading to the gastrointestinal side effects caused by certain non-steroidal anti-inflammatory drugs (like diclofenac or indomethacin) or chemotherapy drugs (such as regorafenib or irinotecan). It involves enterohepatic recirculation and the enzymatic activity of gut microbiome β-glucuronidase overgrowth [55]. In another recent study by Bhatt et al., it was demonstrated that pharmacological intervention on the bacterial β-glucuronidase species proved to be helpful for irinotecan tolerance by reducing the gastrointestinal toxic side effects. The toxicity reduction of irinotecan may be attributed to the incorporation of glucuronic acid to form the glucuronate structure, representing the gastrointestinal disposal form. Under the action of gut microbial β-glucuronidase, the glucuronic acid unit is enzymatically removed, resulting in the production of a product that is harmful to the gastrointestinal mucosa; this is clinically expressed by bloody stools and loss of weight [56].

The GUS may represent a modifiable factor. Additionally, the inhibition of the enzymatic activity of such species could contribute to better chemotherapy outcomes with fewer adverse effects, thereby affecting the efficacy and adherence to such treatment in pancreatic or colorectal cancer patients [55,56].

## 6. Estrogen-Microbiota Gut–Liver Axis

### 6.1. Microbiota in the Gut–Liver Axis

The importance of studying the gut–liver axis was recognized due to the high incidence of liver diseases, in which this axis plays a major role [57]. The gut and liver are linked through the portal vein, biliary tract, and systemic circulation [57,58]. The interconnections appear to function when a proinflammatory medium is set, and the microbial or pathogen-associated molecular patterns (MAMPs/PAMPs) of the gut microbiota are recognized by the receptors of the immune cells of the liver. As a result, an inflammatory process commences and persists, potentially compromising liver function [57].

The liver is deeply involved in the immune response since it is abundant with native immune cells. Kupffer cells represent almost 90% of the whole resident macrophages and, alongside the macrophages descending from monocytes, constitute the initial barrier against intruding microorganisms [59]. Lipopolysaccharides trigger endotoxins from the gut bacteria and play an important role in the hepatic macrophage process of regulation and memory [59] (Figure 1).

Lipopolysaccharides (LPS) represent one of the essential constituents of the Gram-negative bacteria cell envelope and maintain a silent inflammatory state that induces toll-like receptor 4 (TLR-4) stimulation in hepatic immune cells via portal circulation. Subsequently, the inflammatory cascade is activated with the release of interleukin 6 (IL-6), interleukin 1β (IL-1β), and tumor necrosis factor α (TNF-α) [60]. An elevated level of LPS was noticed in patients with fatty liver and diabetes mellitus [60].

The overactivation of liver macrophages by gut lipopolysaccharide toll-like receptor complexes initiates a proinflammatory response associated with lipotoxicity and promotes the production of reactive oxygen species [59].

The rigid layer of the cellular wall of bacteria comprises peptidoglycans (PGN) for Gram-positive and Gram-negative bacteria and lipoteichoic acid (LTA) for Gram-positive bacteria. Both components are ligands for TLR-2, which is also involved in inflammation and hepatic lipotoxic response [60].

The gut-populating microbial species carry specific DNA that is internalized through endocytosis in lysosomal endosomes. These stimulate Toll-like receptors 9 (TLR-9) that activate the inflammatory process via macrophage activation, promoting fatty liver progression [60].

Simultaneously, the liver excretes bile salts and antibacterial agents through the biliary tract into the intestines, which contributes to the physiological process ofeubiosis by regulating gut bacterial overgrowth [58]. When this process is disturbed, i.e., in gut dysbiosis, it can lead to metabolic disorders, ultimately damaging the liver [2,7,58]. Naturally, the grade of liver damage determines the severity of the gut dysbiosis [7,57,58]. The interdependence of this microbiota gut–liver axis cannot be ignored. The immune liver cells influence the intestinal barrier by modifying its permeability through NLRP3 activation and also change the microbiota signature 59. In one of the most extensive prospective studies, which included more than 850 patients (283 with fatty liver), it was demonstrated that a decreased variety of gut microbiota species and an increased population of *Ruminococcus gnavus* were linked to liver steatosis [61].

### 6.2. Estrogen-Microbiota-Gut Axis

The significant role of hepato-biliary-enteric circulation on estrogen metabolism is It is well acknowledged [5]. In fact, C27 cholesterol is the main source of all steroid hormones [2,62], with estrogen being no exception; these hormones are mainly synthesized through the catalyzation of NADPH-dependent cytochrome P450 (CYP) and hydroxysteroid dehydrogenases (HSD) [5,62]. Estrogens are all C18 steroids [2,62]; their aromatic structures consist of one benzene ring, a phenolic hydroxyl group, and a hydroxyl group (17β-estradiol) or a ketone group (estrone) [62]. Estradiol, estrone, and 16-hydroxyestradiol (estriol) are the most abundant endogenous estrogens in the bloodstream [2].

Even if estriol is abundant in the urine of all women (premenopausal/postmenopausal), estradiol presents the highest level of hormonal potency of all estrogens [5,62].

Estrone and estradiol are hydroxylated and converted to catechol estrogens (2-hydroxyestrone, 4-hydroxyestrone, 2-hydroxyestradiol, and 4-hydroxyestradiol) and 16α-hydroxyestrone (estradiol) [62]. From the hydroxylation of estradiol, estriol is formed. Furthermore, the catechol-O-methyltransferase (COMT) enzyme metabolizes or methylates catechol estrogens, forming methoxy-estrogens (2-methoxyestrone, 4-methoxyestrone, 2-methoxyestradiol and 4-methoxyestradiol) [5,62]. Estradiol and 4-hydroxy metabolites are known to have a slight estrogenic activity and carcinogenic ability [5,62].

After the methylation process, in phase II of metabolism, the catalysator UDP-glucuronosyltransferase (UGT) (in GI epithelium or the liver) influences the glucuronidation and methylation process, leading to the attachment of a glucuronic acid moiety and sulfate moiety to numerous endo- and xenobiotics [5,8,16,62]. The conjugated molecules or inactive molecules of estrogen are more hydrophilic and can be dissolved in the blood and then excreted through bile, urine, or feces [2,5]. Several studies have shown that particular estrogen metabolites choose to enter the GI tract through bile to be metabolized further through the deconjugation process performed by gm-GUS [5,16].

## 7. The Estrobolome and Hepatocellular Adenomas

This metabolic pathway could represent the missing link of HCA occurrence when no additional risk factors are present. Around 10–20% of HCA are described in males, most of whom have no identifiable risk factors. Noticeably, HCA in males is considered to present a high risk for malignant transformation [63,64]. Another consideration is the fact that the occurrence of HCA is linked to oral contraceptive usage in women, but not all women exposed to this hormonal therapy are diagnosed with HCA. Moreover, it has been stated that there is a relationship between liver steatosis and hepatic adenomas, but the important role of the gut microbiota in hepatic steatosis should also be considered [43]. Estrogens are linked to increased Sig-R1 expression due to HNF1α deprivation, the consequence of which is hepatic cell overgrowth and hepatic fatty transformation, appearing as a similar process in this type of hepatic adenoma [43]. Villemain suggested that it would be worth testing the Sig-R1 ligands (that are already used in neuro-psychiatric diseases) in cases of hepatic tumoral proliferation or with liver lipogenesis [43].

We suggest that a specific signature or imbalanced estrobolome should be considered and studied to elucidate some previously overlooked factors.

HCA has an increasing incidence, and the traditional outline of individuals with HCA is changing due to the identification of novel risk factors [65]. The estrobolome plays a prominent role in encoding and mediating estrogen metabolism in the GI tract. Therefore, it begins the process of the GUS reverse of glucuronidation of estrogen-conjugated molecules and cleavage of the glucuronic moiety from estrogen, ultimately resulting in the reactivation of estrogen. This process facilitates the reabsorption of active estrogen into the bloodstream through the lining of intestinal mucosa via the portal vein. In terms of novel and future approaches and manipulation of microbiota, the concept of pharmaco-microbiomics is evolving; it comprises the complex study of microbiome interactions with the bioavailability, toxic adverse effects, and bioactivity of various drugs. The perspective of pharmaco-microbiomics addresses genomics, pharmacologic, and microbiologic aspects of this special relationship between medication and gut microbiota [55].

All of the above factors explain the mechanism of excessive exposure to estrogen (endogenous or exogenous) and estrogen metabolism via the reactivation of conjugated estrogen by GUS, leading to the absorption of unconjugated estrogen into the blood flow and the continuous vicious circle that may lead to numerous estrogen-related pathologies, including HCA. Although the relationship between estrogen exposure and HCA has been well established, the mechanisms are not completely understood, particularly in cases with no identifiable cause. In such cases, we recommend considering the role of an estrobolome imbalance, specifically, the perturbance of bacterial species possessing β-glucuronidase, as the overlooked metabolic link. Microbiota and the search for a specific signature of the estrobolome should be the subject of future studies to connect the dots.

## Figures and Tables

**Figure 1 ijms-24-16034-f001:**
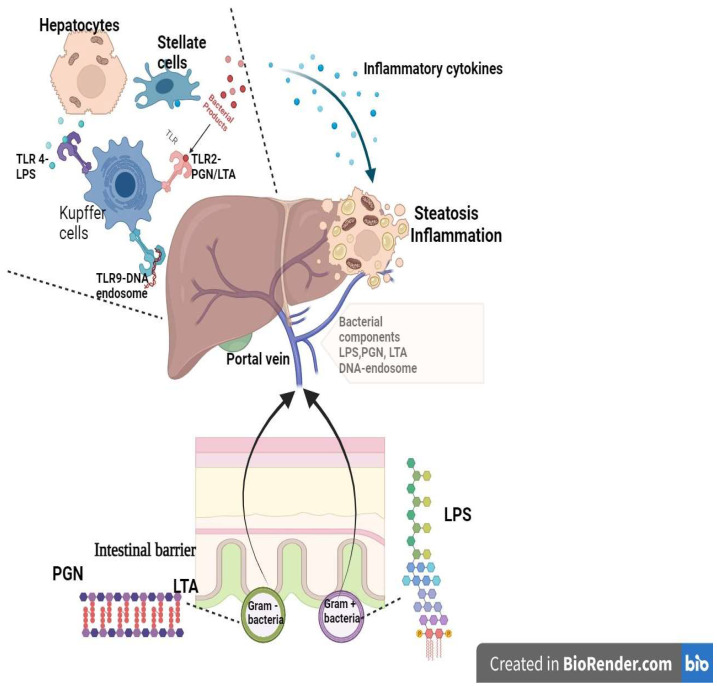
Gut bacterial triggers and the liver immunological response (created in BioRender.com). The components of the gut bacterial envelope are triggers for the hepatic immune response. The LPS are the components of the external layer of the wall of Gram-negative bacteria, while PGN are the components of the rigid layer of Gram-positive bacteria (alongside LTA) and of the internal wall of Gram-negative bacteria. The LPS, PGN, LTA, and bacterial DNA-endosome enter into the portal circulation, bypassing the intestinal barrier. In the liver, these components are ligands for TLR receptors and promote the pro-inflammatory response by macrophage activation and cytokine release. The consequence is hepatic inflammation and steatosis. TLR—Toll-like receptor; PGN—peptidoglycan; LTA—lipoteichoic acid; LPS—lipopolysaccharide.

## Data Availability

No new data were created or analyzed in this study. Data sharing is not applicable to this article.

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
