# Peer review of "Estrobolome and Hepatocellular Adenomas—Connecting the Dots of the Gut Microbial β-Glucuronidase Pathway as a Metabolic Link"

_ijms, 2023, doi:10.3390/ijms242216034_

Round 1
Reviewer 1 Report
Comments and Suggestions for Authors
In this review article, authors have described a potential link between estrogen metabolism and gut microbiome in development of hepatocellular adenoma (HCA). HCA is relatively rare type of liver cancer, associated with usage of controceptives, and less studies. Overall, the manuscript is well written and interest to the field. Here are some specific points that can be considered to be incorprated into the review:
1. There may not be necessary to have methodology for review articles.
2. The authors claimed that gut microbiome may play a role in HCA. Are there any evidence of specific microbiome that associate with development of HCA? Particularly the species with estrogen metabolizing activities?
3. The authors pointed out the importance of β-glucuronidase expressing bacteria to convert estrogen into estradiol glucuronide. However, could estradiol glucuronide promote HCA or HCC formation? What is the possible mechanism?
4. How estrogen metabolite formed by gut microbiome is transferred back to blood stream. Is it by active transport of intestinal epithelial cells? or absorbed with bile acids and other lipids?
5. How the link between estrogen-microbiome axis can be intervened for treatment of HCA?
Author Response
Dear Reviewer,
We want to thank you for taking the time to assess our manuscript and appreciate effort that you dedicated to provide feedback on our manuscript. We carefully considered your questions and we addressed everyone of them and highlighted incorporated into the manuscript.
In this review article, authors have described a potential link between estrogen metabolism and gut microbiome in development of hepatocellular adenoma (HCA). HCA is relatively rare type of liver cancer, associated with usage of contraceptives, and less studies. Overall, the manuscript is well written and interest to the field. Here are some specific points that can be considered to be incorporated into the review:
- There may not be necessary to have methodology for review articles.
Response: Thank you for your suggestion, we considered that it should be mentioned, although it is a scoping review, but indeed it might not be mandatory.
- The authors claimed that gut microbiome may play a role in HCA. Are there any evidence of specific microbiome that associate with development of HCA? Particularly the species with estrogen metabolizing activities?
Response: As you well pointed, this was the purpose of our review – to highlight those missing links between microbiota and HCA. Since there are no reviews on this topic and few papers to address this issue and we aim to provide this new perspective, since HCA are close related to estrogen metabolism and receptors.
- The authors pointed out the importance of β-glucuronidase expressing bacteria to convert estrogen into estradiol glucuronide. However, could estradiol glucuronide promote HCA or HCC formation? What is the possible mechanism?
Response: The mechanisms of hepatic cells proliferation is through estrogen receptor - ER-α . The estrogen glucuronide is activated to estrogen by glucuronidase gut bacterial activity, by removing the glucuronic acid unit. This estrogen activity is ER-α mediated at transcriptomic stage and this receptor is the target for estrogen signaling and liganded ER-α exerts control over the transcription of genes specific to its binding. In the preclinical study of Guillaume et al. it was described the degree of liver ER-α involvement in metabolic balance, since ER-α was considered a biomolecule of interest for liver steatosis and this study concluded that the specific regulation of these hepatic estrogen receptors could prevent dysmetabolic disorders. Consistently Sig-R1 was found to participate in fatty liver disease and may modify the hepatocellular adenoma phenotype, since it is also a regulator of hepatic cells proliferation. The potential pathways of HCA evolution to malignancy involves JAK-STAT pathway modulation and activation and in animal studies there were describes inhibition of cyclin D and estrogen receptor α36 (ERα36) super-expression. In the study by Lau et al. it was suggested a new connection between endoplasmic reticulum stress and impaired cyclin D and ER-α36 in the evolution of HCA to malignancy. Additionally, in HCA was found an inhibited activity of HNF1α associated with an overexpression of Sigma receptor 1 and the possible explanation was that HCA could evolve from a different cellular line than HCC.
- How estrogen metabolite formed by gut microbiome is transferred back to blood stream. Is it by active transport of intestinal epithelial cells? or absorbed with bile acids and other lipids?
Response: Estradiol and estrone are lipophilic, but with enough water solubility properties, although the sulfates and glucuronide estrogen form have a high water solubility. The gut microbiome deconjugates bile acid-secreted conjugated estrogen via the bacterial secretion of beta- glucuronidase, allowing the transformed estrogen to be absorbed by the intestine and enter in the blood stream, in the same way as oral estrogens. Also, dysbiosis favorize the gut-barrier increased permeability and the passage into the blood stream. The deconjugated estrogens from the gut are reabsorbed into the enterohepatic circulation, by easily diffuse through the cells membranes.
- How the link between estrogen-microbiome axis can be intervened for treatment of HCA?
Response: This is a beautiful question to be answered because the strategies that are already proved to be useful as interventions on β-glucuronide microbiota should be tried in managing the microbiota of this patients. In the study by Bhatt et al. it was demonstrated that the pharmacological intervention on bacterial β- glucuronidase species proved to be helpful in preventing the removal of the glucuronic acid unit, which has an anti-toxic and protective role. In terms of novel and future approach and manipulation of microbiota it is developing the concept of pharmacomicrobiomics that comprises the complex study of microbiome interaction with drugs bioavailability, toxic adverse effects and bioactivity. The perspective of pharmacomicrobiomics addresses to genomics, pharmacologic and microbiologic aspects of this special relationship between medication and gut microbiota influence.
We look forward to hearing from you in due time regarding our submission and to respond to any further questions and comments you may have.
Sincerely,
Sandica Bucurica
Reviewer 2 Report
Comments and Suggestions for Authors
This study is important and will certainly be helpful to physicians, other caregivers and scientists studying the pathogenesis and natural history of hepatic adenomas. You point out that males with hepatocellular adenomas are particularly predisposed to hepatocellular carcinoma but often it is difficult to differentiate benign from malignant cases. You allude to focusing on pharmacomicrobiomics. At this point in time what would you recommend, other than resection of the lesion, would be a reasonable screening test on the initial encounter with the male patient? Any studies on the estrobolomes?
Furthermore your studies are in Romania and as I recall there used to be a gypsy population that is reluctant to be studied for anything. Do you have any data on the prevalence of hepatic adenomas in gypsy men? If so benign vs malignant?
Author Response
Dear reviewer,
We appreciate the time and effort that you dedicated to provide feedback on our manuscript and we are grateful for the interesting comments, which we carefully addressed, and highlighted in the manuscript.
This study is important and will certainly be helpful to physicians, other caregivers and scientists studying the pathogenesis and natural history of hepatic adenomas. You point out that males with hepatocellular adenomas are particularly predisposed to hepatocellular carcinoma but often it is difficult to differentiate benign from malignant cases.
At this point in time what would you recommend, other than resection of the lesion, would be a reasonable screening test on the initial encounter with the male patient? Any studies on the estrobolomes?
Response: Male sex represents a high risk factor for oncogenesis of HCA, alongside the β-catenin exon 3 mutation, and the substantial diagnostic difficulties are for the smaller lesions. The magnetic resonance exams with specific contrast agents were indicated for the possibility of discriminating malignancy in early stages. The future studies are mandatory for establishing male patients estrobolome signature.
Furthermore your studies are in Romania and as I recall there used to be a gypsy population that is reluctant to be studied for anything. Do you have any data on the prevalence of hepatic adenomas in gypsy men? If so benign vs malignant?
Response: Unfortunately, we do not posses data regarding the incidence of hepatic adenoma in rroma ethnic population group.